rsob.royalsocietypublishing.org

Autophagy in crop plants: what's new beyond
*Arabidopsis*? *Open Biol.* **8**: 180162.

Subject Area:
cellular biology

Keywords:
autophagy, crops, development,
plant–microbe interactions, stress responses

Author for correspondence:
Diane C. Bassham
e-mail: bassham@iastate.edu

# Autophagy in crop plants: what's new beyond *Arabidopsis*?

Jie Tang and Diane C. Bassham

Department of Genetics, Development and Cell Biology, Iowa State University, Ames, IA 50011, USA

 JT, 0000-0003-3253-4894; DCB, 0000-0001-7411-9360

Autophagy is a major degradation and recycling pathway in plants. It functions to maintain cellular homeostasis and is induced by environmental cues and developmental stimuli. Over the past decade, the study of autophagy has expanded from model plants to crop species. Many features of the core machinery and physiological functions of autophagy are conserved among diverse organisms. However, several novel functions and regulators of autophagy have been characterized in individual plant species. In light of its critical role in development and stress responses, a better understanding of autophagy in crop plants may eventually lead to beneficial agricultural applications. Here, we review recent progress on understanding autophagy in crops and discuss potential future research directions.

## 1. Introduction

Autophagy (meaning 'self-eating') is a process conserved throughout eukaryotes for degradation and recycling of cytoplasmic components (proteins, protein aggregates and organelles) during development or environmental stress. In plants, two major types of autophagy, microautophagy and macroautophagy, have been described [1]. During microautophagy, cytoplasmic components are taken up by the vacuole through the invagination of the tonoplast. Anthocyanin and membrane-damaged chloroplasts have been shown to be taken up by the vacuole through microautophagy in *Arabidopsis* [2,3] but little is known about the factors required for this process in plants. On the other hand, when macroautophagy is activated, a cup-shaped membrane structure named a phagophore forms around the cargo, expands and finally seals as a double-membrane vesicle called an autophagosome, of which the outer membrane eventually fuses with the tonoplast and releases an autophagic body into the vacuole for degradation [1,4].

Macroautophagy is the best-characterized type of autophagy in plants and other organisms, and is therefore often simply referred to as autophagy. The core autophagic machinery consists of autophagy-related (ATG) proteins, which function during the induction of autophagy and formation of autophagosomes. ATG proteins can be divided into four major groups. The ATG1/ATG13 kinase complex responds to upstream signals and induces downstream autophagosome formation [5]. The only transmembrane core ATG protein, ATG9, has been proposed to deliver membrane to the phagophore for autophagosome formation [6] and was recently identified as a key player in endoplasmic reticulum (ER)-derived autophagosome formation in *Arabidopsis* [7]. The phosphatidylinositol-3-kinase (PI3K) complex is essential for phosphorylating phosphatidylinositol to produce phosphatidylinositol-3-phosphate (PI3P), which is required to recruit proteins involved in autophagy [4]. For example, in *Arabidopsis*, SH3 domain-containing protein 2 (SH3P2) is a recently identified protein that binds to PI3P and is potentially involved in autophagosome membrane remodelling and autophagosome fusion with the vacuole [6,8]. ATG8 conjugation to the membrane lipid phosphatidylethanolamine (PE) requires two ubiquitin conjugation-like pathways, the ATG5–ATG12 and ATG8–PE

**Table 1.** Number of core *ATG* genes in selected species.

| | *Arabidopsis* | barley | grapevine | maize | rice | tobacco | tomato |
|---|---|---|---|---|---|---|---|
| ATG1 | 4 | 2 | 2 | 4 | 3 | 3 | 2 |
| ATG2 | 1 | 2 | 1 | 1 | 1 | 1 | 1 |
| ATG3 | 1 | 1 | 0 | 1 | 2 | 1 | 1 |
| ATG4 | 2 | 1 | 1 | 2 | 2 | 1 | 1 |
| ATG5 | 1 | 1 | 1 | 1 | 1 | 1 | 2 |
| ATG6 | 1 | 1 | 1 | 2 | 3 | 1 | 1 |
| ATG7 | 1 | 1 | 1 | 1 | 1 | 1 | 1 |
| ATG8 | 9 | 3 | 6 | 5 | 7 | 5 | 7 |
| ATG9 | 1 | 1 | 1 | 1 | 2 | 1 | 1 |
| ATG10 | 1 | 1 | 1 | 1 | 2 | 1 | 1 |
| ATG11 | 1 | 1 | 1 | 2 | 1 | 2 | *1* |
| ATG12 | 2 | 1 | 1 | 1 | 1 | 3 | 1 |
| ATG13 | 2 | 1 | 2 | 6 | 2 | 3 | 2 |
| ATG14 | 2 | *1*[a] | *1* | 2 | *1* | 2 | *1* |
| ATG16L | 1 | 1 | *1* | 1 | 1 | 2 | *1* |
| ATG18 | 8 | 5 | 7 | 10 | 6 | 6 | 6 |
| References | [9,13,26,27] | [18] | [14] | [13] | [14,20] | [14,19] | [22,23] |

[a]Italic numbers indicate that the gene of the corresponding species is not included in published papers. BLASTP [28] was used to identify these genes using protein sequences from *Arabidopsis*, using http://www.gramene.org/ (barley (Hv_IBSC_PGSB_v2), grapevine (IGGP_12x), maize (AGPv4)), https://rapdb.dna.affrc.go.jp/tools/blast/ (rice (IRGSP-1.0)) or https://solgenomics.net/ (tobacco (Nitab v.4.5 proteins Edwards2017) and tomato (ITAG release 3.20)) with default parameters. Detailed information on newly identified genes is included in table 2.

conjugation systems. This lipid attachment anchors ATG8 to the expanding phagophore during formation of autophagosomes and allows its function as a docking site for autophagic cargo receptors [4].

In plants, autophagy has been shown to function in response to various environmental stresses such as nutrient starvation, drought, salt and heat [9–11]. Autophagy-defective mutants are more sensitive to stress, indicating that autophagy contributes to plant survival under stress conditions. In addition, *atg* mutants in both maize and *Arabidopsis* produce fewer seeds compared to wild-type (WT) plants, while *ATG*-overexpressing *Arabidopsis* plants have increased autophagy, and higher seed production and oil accumulation [12,13], suggesting that autophagy also functions during seed development.

The expanding global population and environmental challenges will increasingly contribute to food insecurity for large segments of the population [1]. The functions of autophagy in stress responses and seed production suggest a potential for autophagy as a target for manipulation that may lead to agronomic benefits. In this review, we summarize and discuss the current understanding of autophagy in crop plants, focusing on its functions in development, stress responses and plant–microbe interactions.

## 2. Identification and characterization of *ATG* genes

Considering the important roles of *ATG* genes during autophagy, identifying *ATG* loci in the genomes of crop species and characterizing their functions is essential in expanding our understanding of autophagy in these species. To date, core *ATG* genes have been identified in at least 14 crop species. Sequences of *ATG* genes (or ATG proteins) from *Arabidopsis* and rice were used as queries to search against corresponding genomic or protein sequences for most of these species [13–25]. In general, most of the core *ATG* genes can be identified in each species (tables 1 and 2). No homologues of ATG3 have been found in grapevine. However, whether this is due to incomplete genome sequence information needs further confirmation. As in *Arabidopsis*, ATG proteins in crops are typically encoded by a single gene or a small gene family. It appears that ATG1, ATG8 and ATG18 are encoded by small gene families in all of these species (table 1). The number of other *ATG* gene family members varies among species.

Autophagy-defective plants have been characterized in several species. In *Arabidopsis*, mutants in various *ATG* genes show characteristic phenotypes such as reduced growth, early senescence, decreased seed production and hypersensitivity to abiotic stress [29]. The phenotypes of *atg* mutants under biotic stress are more complicated; mutants can be more sensitive or more resistant to pathogens depending on the lifestyle of the pathogen and the age-related salicylic acid (SA) levels in the plant [30,31]. Similar phenotypes have been observed in various crop species. Maize *atg12* mutants show early senescence, hypersensitivity to nitrogen starvation and decreased yield [13]. A barley RNA interference (RNAi)-*ATG6* line is sensitive to carbon starvation and oxidative stress [32]. Rice *atg7* and *atg9* mutants also show reduced vegetative growth and early senescence [33]. However, they are unable to produce seeds due to male sterility [34], an additional phenotype that is not

rsob.royalsocietypublishing.org    Open Biol. 8: 180162

**Table 2.** Information on newly identified autophagy genes in crop species.

| protein | *Arabidopsis* gene ID | crop species | crop gene ID | identity/similarity to *Arabidopsis* protein[a] |
|---|---|---|---|---|
| ATG11 | At4g30790 | tomato | Solyc07g005970 | 56%/74% |
| ATG14a[b] | At1g77890 | barley | HORVU2Hr1G029220 | 40%/60% |
| | | grapevine | VIT_18s0001g04540 | 52%/67% |
| | | maize | Zm00001d022199 | 40%/57% |
| | | | Zm00001d006916 | 38%/56% |
| | | rice | Os07g0626300 | 42%/58% |
| | | tobacco | Nitab4.5_0004952g0050 | 50%/66% |
| | | | Nitab4.5_0001915g0140 | 49%/66% |
| | | tomato | Solyc04g072440 | 40%/55% |
| ATG14b[b] | At4g08540 | barley | HORVU2Hr1G029220 | 48%/64% |
| | | grapevine | VIT_18s0001g04540 | 70%/83% |
| | | maize | Zm00001d022199 | 47%/61% |
| | | | Zm00001d006916 | 46%/61% |
| | | rice | Os07g0626300 | 52%/67% |
| | | tobacco | Nitab4.5_0001915g0140 | 67%/81% |
| | | | Nitab4.5_0004952g0050 | 68%/81% |
| | | tomato | Solyc04g072440 | 55%/67% |
| ATG16L | At5g50230 | grapevine | VIT_17s0000g09750 | 69%/86% |
| | | tobacco | Nitab4.5_0000510g0020 | 70%/85% |
| | | | Nitab4.5_0001812g0090 | 70%/85% |
| | | tomato | Solyc03g111740 | 71%/85% |

[a]Per cent identity and similarity was determined for the predicted protein sequence using BLASTP [28].
[b]Searches for ATG14a and ATG14b identified the same proteins in the listed species. Here, we list per cent identity and similarity of these proteins compared to both AtATG14a and AtATG14b.

observed in *Arabidopsis* or in maize. These results suggest that autophagy may have conserved functions among distinct species but may also perform species-specific roles.

Numerous studies have now been performed on autophagy in crops, revealing functions of autophagy in stress responses, development, hormone responses, metabolism and cell death (table 3). These functions will be discussed in more detail in the following sections.

# 3. Functions of autophagy during development

## 3.1. Leaf senescence

Leaf senescence is considered to be an important developmental process because of its critical role in remobilizing nutrients from mature leaves to support developing organs (e.g. developing seeds) [87]. It is logical to hypothesize that autophagy functions during leaf senescence because autophagy is a degradation and recycling process. However, somewhat contradictory to this idea is the observation of early senescence in various *Arabidopsis atg* mutants [9,88], a rice *atg7* mutant [33] and a maize *atg12* mutant [13]. A proposed explanation for this phenotype is that the down-regulation of flavonoid biosynthesis in *atg* mutants causes oxidative stress, which further triggers the biosynthesis and accumulation of excess SA, leading to leaf yellowing

[31,89]. The involvement of autophagy in senescence is supported by the observation that many *ATG* transcripts are upregulated in older leaves. For example, in *Arabidopsis*, 15 *ATG* genes are upregulated during senescence [90]. The increased expression of *ATG* genes during senescence has also been shown in apple [35–37], barley [18,40] and soybean [61]. Similarly, 30 and 27 *ATG* genes were shown to be upregulated in older leaves and the leaf tip (the oldest part of a leaf), respectively, in maize [13]. An additional piece of evidence for the involvement of autophagy during senescence comes from the higher accumulation of lipidated ATG8, which represents a higher level of autophagy activity, in the yellowing area of a senescing leaf compared to the green area of the same leaf in maize [21].

Nitrogen (N) is one of the predominantly remobilized elements during senescence [91]. Remobilized N contributes 45% of the total N in new rice leaves and 50% to 90% of grain-filling N in cereals including rice, wheat and maize [92]. A useful method to monitor N remobilization is $^{15}$N tracing, in which a short-term $^{15}$N treatment is applied to plants at the vegetative stage. At harvest, dry weight, amount of total N and amount of $^{15}$N are measured in seeds and various vegetative tissues, including leaves, roots and stems, to calculate indices such as N use efficiency (NUE), N remobilization efficiency (NRE) and N harvest index (NHI) [91]. The role of autophagy during N remobilization was first studied in *Arabidopsis* by applying this method to WT plants and autophagy-defective mutants [93]. It was shown that the

**Table 3.** Crop species with identified *ATG* genes and potential processes that require autophagy.

| species | related processes and references |
| --- | --- |
| apple (*Malus domestica*) | vegetative growth [35,36], senescence [35–37], abiotic stress response [35–39], metabolism [38] |
| banana (*Musa acuminata*) | biotic stress response [15], hormone response [15], cell death [15] |
| barley (*Hordeum vulgare*) | senescence [18,40], nutrient remobilization [40], microspore embryogenesis [41], abiotic and biotic stress response [18,32,41,42], cell death [41,42] |
| cassava (*Manihot esculenta*) | biotic stress response [24,43] |
| common bean (*Phaseolus vulgaris*) | nodule development [44–46] |
| foxtail millet (*Setaria italica*) | abiotic stress response [17] |
| grapevine (*Vitis vinifera*) | abiotic stress response [14], fruit ripening [47] |
| maize (*Zea mays*) | senescence [13,21], nutrient remobilization [13], seed development [13,21,48], abiotic stress response [13,21,48,49] |
| pepper (*Capsicum annuum*) | abiotic stress response [16] |
| rice (*Oryza sativa*) | vegetative growth [33], senescence [33,50–52], nutrient remobilization [33], anther development [34,53], metabolism [34,53], abiotic and biotic stress response [20,50,54–60], cell death [34,52,55,56] |
| soybean (*Glycine max*) | nutrient remobilization [61], abiotic stress response [62] |
| tobacco (*Nicotiana tabacum*) | abiotic and biotic stress response [19,63,64], hormone response [19] |
| tomato (*Solanum lycopersicum*) | anther development [65], abiotic and biotic stress response [22,23,66–75], hormone response [66], cell death [65,66,68,69,71,76] |
| wheat (*Triticum aestivum*) | phloem development [77,78], spikelet development [79], seed development [80], abiotic and biotic stress response [25,81–86] |

accumulation of labelled N in seeds and NRE were both significantly lower in autophagy-defective mutants compared to WT [93]. This approach has also been used in maize and rice, and defects in autophagy suppressed N remobilization [13,33]. These results indicate that autophagy is required for efficient N remobilization during leaf senescence and seed set.

## 3.2. Seed development

Autophagy impacts seed production not only owing to its important role during senescence and nutrient remobilization but also by functioning during seed development. A recent study in *Arabidopsis* revealed that almost all of the *ATG* genes were upregulated in siliques during seed development [94]. In maize, increased transcript abundance of many *ATG* genes was also observed in the endosperm but not the embryo [13]. Additionally, an ATG8 lipidation assay showed that the accumulation of ATG8–PE adducts in maize endosperm started at 18 days after pollination (DAP) and increased until the last time point used in the experiment, 30 DAP, indicating that autophagy was activated during endosperm development [21].

Autophagy may contribute to the transport of seed storage proteins. In wheat, electron microscopy showed that prolamins were transported from the ER to protein storage vacuoles (PSVs) through an autophagy-like pathway [80], although whether this involves *ATG* genes is unclear. In the seeds of *Arabidopsis atg5*, *atg7* and *atg4a atg4b* mutants, a decrease in the amount of 12S globulins and increased amounts of 12S globulin precursors were observed compared to WT, again suggesting that autophagy might be involved in

delivering the precursors to the PSVs, the site of processing of the precursors to the mature form [94].

## 3.3. Reproductive development

Although *Arabidopsis atg* mutants have reduced fecundity, they can still complete normal life cycles and produce viable seeds [95]. One exception is the *Arabidopsis atg6* mutant, which exhibits a pollen germination deficiency [96–98]. However, defects in autophagy may not contribute to this phenotype, as ATG6 and the PI3K complex are involved in multiple biological processes [96]. The first piece of direct evidence connecting autophagy to reproductive development was found in wheat. During wheat floret development, many floret primordia undergo abortion rather than reaching the fertile floret stage, a phenomenon that is enhanced by long-day conditions [79]. In the aborting florets, autophagy was found to occur in the ovary cells undergoing programmed cell death (PCD). Transmission electron microscopy (TEM) indicated the formation of double-membrane vesicles that finally fused with the vacuole, releasing single-membrane structures into the vacuole [79]. In addition, *ATG4* and *ATG8* were upregulated during this process. Combining these results with other findings including decreased soluble carbohydrates in the spikes, the researchers proposed a model for floret abortion in which under long-day conditions, accelerated growth increases the consumption of carbohydrates, leading to a starvation condition that triggers autophagy-mediated programmed cell death [79].

On the other hand, characterization of rice *atg7* and *atg9* mutants relates autophagy to anther development. The two mutants exhibit a striking male sterility phenotype [34].

Further analyses were conducted in the *atg7* mutant to characterize this phenotype. TEM analysis detected autophagosome-like structures in the tapetum cells at the uninucleate stage of anther development, while no autophagosome-like structures were observed at this stage in the *atg7* mutant. The tapetum is degraded by PCD to supply nutrients during pollen development [99]. The tapetum was fully degraded in WT but only partially degraded in the *atg7* mutant; lipid bodies, plastids and mitochondria were found remaining in the cytoplasm, suggesting that autophagy is required for degradation of cellular components during pollen development [34]. Lipid profiling of pollen further showed that lipid metabolism was altered in the autophagy-defective mutant. For example, triacylglycerol, a major component of pollen lipid bodies that is important for pollen maturation, was decreased in the *atg7* mutant [34]. In addition, levels of bioactive gibberellin (GA) and cytokinin were decreased in *atg7*. Application of exogenous GA to *atg7* recovered pollen maturation and partially recovered pollen germination, but the male sterility phenotype was not rescued [53], indicating that altered hormone levels contribute to the phenotypes of the *atg7* mutant.

Besides the development of functional pollen, pollen release is another critical process for male fertility [100]. The last step of pollen release is dehiscence, during which the stomium, a furrow separating anther lobules, breaks [101,102]. Around the stomium are epidermal cells, which undergo PCD during development, a process required for stomium breakage [103,104]. In tomato, autophagic vesicles were observed in the epidermal cells surrounding the stomium during their PCD, suggesting that autophagy is involved in this process [65].

## 3.4. Vascular development

Vascular tissues are important for plants due to their critical functions in mechanical support and long-distance transport [105]. Xylem and phloem are the two major conductive tissues, and consist of cell types highly specialized for this function [106]. The main conductive cells in xylem are tracheary elements (TEs), which undergo PCD and clear their cellular contents completely during differentiation [106]. A role for autophagy in xylem development was first demonstrated in poplar [107]. Upregulation of several *ATG* genes was observed during the PCD of poplar xylem fibre cells, and autophagy was also observed in fibrous and pioneer roots under field conditions [108,109]. Autophagy also functions in TE differentiation in *Arabidopsis* [110], evidenced by a decreased xylem cell number in an *atg5* mutant compared with WT, upregulation of several *ATG* genes during TE differentiation, and the presence of autophagosome-like structures in differentiating cells. The small GTP-binding protein RabG3b, a protein co-localized with ATG8, was shown to be a positive regulator of autophagy and TE differentiation [110]. Heterologous overexpression of *Arabidopsis RabG3b* in poplar enhanced xylem development and growth rate [111]. In addition, METACASPASE9 (MC9) was recently identified as a negative regulator of autophagy during TE differentiation and is thought to restrict autophagic cell death to the target cells [112].

Sieve elements (SEs) are responsible for transport in phloem. Unlike TEs, SEs undergo only partial degradation of cell organelles and retain others during their differentiation [106]. A recent study indicated that microautophagy may be involved in SE differentiation in wheat [77,78]. TEM imaging showed that cytoplasm entered the vacuole through the invagination of the tonoplast during the process of PCD, presumably leading to degradation of cytoplasmic content [77].

# 4. Functions of autophagy during abiotic stress

## 4.1. Nutrient starvation

Autophagy has been extensively studied as an abiotic stress response, and one typical stress condition that induces autophagy is nutrient starvation. Besides *Arabidopsis*, autophagy was shown to function in responses to nutrient deprivation (carbon or N) in apple [35–38], barley [18,32], foxtail millet [17], grapevine [14], maize [13,21], pepper [16], rice [20,50] and wheat [81], as indicated by the upregulation of *ATG* genes or the hypersensitivity of autophagy-defective mutants to starvation. Recently, a study illustrated that the overexpression of *ATG18a* in apple conferred increased tolerance to N starvation and led to increased autophagy under these conditions [38]. Compared to WT plants under N depletion conditions, several pathways and their corresponding genes were further upregulated in response to N depletion in *ATG18a* overexpressing plants. First, anthocyanin biosynthesis was upregulated and anthocyanin accumulation was higher [38]. Stress conditions trigger reactive oxygen species (ROS) production [113], and it would be reasonable to hypothesize that, as an antioxidant, anthocyanin may help prevent damage by ROS, contributing to higher stress tolerance. Another important upregulated pathway was nitrate absorption and assimilation, including several high-affinity nitrate transporters and a nitrate reductase that is required for nitrate assimilation. Increased nitrate was found in overexpressing plants compared to WT under N starvation [38]; this might be a key factor contributing to higher tolerance of overexpressing plants to N deficiency. Other upregulated genes or pathways included *ATG8i*, *ATG9* and the starch degradation pathway. However, how the overexpression of *ATG18a* is connected to the upregulation of other pathways is still unknown.

## 4.2. Drought stress

Drought is another common environmental stress that plants may encounter. The involvement of autophagy in the response to drought stress was first elucidated in *Arabidopsis*, indicated by the upregulation of *ATG18a* and the induction of autophagosome formation by osmotic stress [10]. In crops, various studies have shown the upregulation of *ATG* genes in response to drought stress, for example, in apple [37,38], barley [32], foxtail millet [17], pepper [16], rice [20], tomato [22,66] and wheat [81–83]. In *Arabidopsis*, *atg5*, *atg7* and RNAi-*ATG18a* mutants, which are unable to activate autophagy during drought, are hypersensitive to drought stress [10,114], suggesting that autophagy is important for plant survival in drought conditions. Similar phenotypes were also observed in autophagy-defective tomato and wheat plants [22,66,83]. In addition, overexpression of *ATG18a* in apple conferred higher autophagy activity and increased drought tolerance, further supporting a key role for autophagy in responses to drought [39].

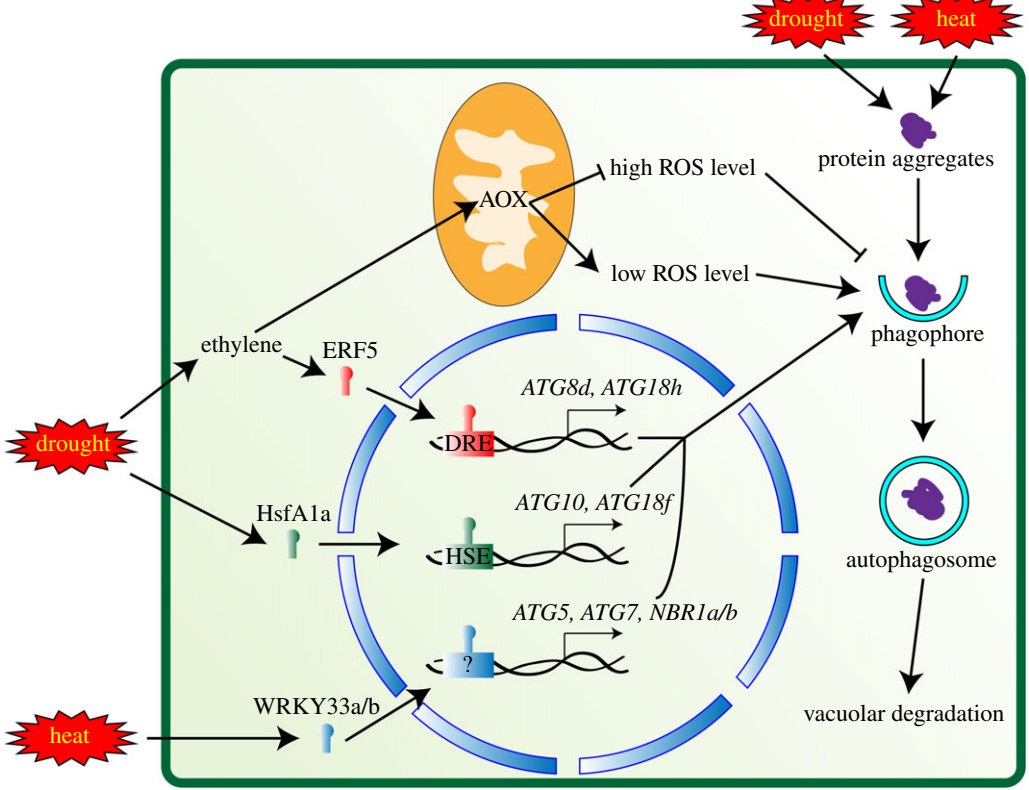

**Figure 1.** Identified regulators of autophagy during drought and heat stress in tomato. In tomato, alternative oxidase (AOX) within mitochondria and the transcription factor ethylene response factor 5 (ERF5) are induced by drought stress, in a process mediated by ethylene. AOX can positively regulate autophagy by balancing the level of reactive oxygen species (ROS); lower ROS levels are thought to activate autophagy, whereas higher ROS levels inhibit autophagy. ERF5 induces the expression of *ATG8d* and *ATG18h* by binding to drought-responsive elements (DRE) in their promoters. Heat-shock transcription factor A1a (HsfA1a) is also induced by drought stress and activates the expression of *ATG10* and *ATG18f* by binding to heat-shock elements (HSE) in their promoters. Under heat stress, the transcription factors WRKY33a and WRKY33b activate the expression of *ATG5*, *ATG7*, *NBR1a* and *NBR1b* in tomato. Autophagy in turn functions to degrade the protein aggregates induced by drought or heat.

Abiotic and biotic stresses can induce the production of ROS in plants as signalling components for defence responses [113,115]. One major source of ROS is plasma membrane-associated NADPH oxidase [116]. In *Arabidopsis*, inhibition of NADPH oxidase blocks activation of autophagy under salt stress and nutrient starvation conditions but not under osmotic stress, suggesting that NADPH oxidase-mediated ROS signalling is necessary for starvation- and salt-induced autophagy but not for drought-induced autophagy [10]. A recent study in tomato suggested that mitochondrial alternative oxidase (AOX) may regulate autophagy through mitochondrial ROS during drought stress [66]. AOX limits the formation of ROS by preventing over-reduction of the electron transport chain [117]. In tomato, *AOX*-overexpressing plants showed increased tolerance to drought, while *AOX*-silenced plants showed hypersensitivity compared to WT, suggesting that AOX functions in drought responses [66]. Overexpression of *AOX* led to higher autophagy activity in drought, while lower autophagy activity was observed upon silencing of *AOX*. Application of exogenous $H_2O_2$ to *AOX*-overexpressing plants decreased autophagy activity and application of exogenous dimethyl thiourea (an $H_2O_2$ scavenger) increased autophagy activity in *AOX*-silenced plants, suggesting that AOX can regulate autophagy activity by changing ROS levels [66]. Additionally, the plant hormone ethylene (ET) is involved in this process. AOX transcript and protein was induced by treatment with the ET precursor 1-(aminocarbonyl)-1cyclopropanecarboxylic acid (ACC).

ACC treatment conferred higher autophagy activity and better drought tolerance to plants either overexpressing or silencing *AOX* [66] (figure 1).

*ATG* genes are transcriptionally regulated by drought in tomato. The transcription factor ethylene response factor 5 (ERF5), induced by both drought stress and ACC treatment, binds to the promoters of *ATG8d* and *ATG18h*, inducing their expression [66]. Another transcription factor, heat-shock transcription factor A1a (HsfA1a) [22] is induced by drought stress and functions as a positive regulator of drought responses. The induction of autophagy by drought stress is higher in *HsfA1a*-overexpressing plants and lower in *HsfA1a*-silenced plants, as indicated by the number of autophagosomes and the level of ATG8 lipidation. HsfA1a was found to bind to the heat-shock element (HSE) in the promoters of *ATG10* and *ATG18f* and induce their expression [22] (figure 1).

Although ROS are important for autophagy activation, the accumulation of ROS is also toxic to cells, causing membrane disruption, protein aggregation and even cell death [113,115]. One proposed function of autophagy during drought stress is to control protein quality. In tomato, silencing *HsfA1a* led to higher accumulation of insoluble proteins, while overexpression of *HsfA1a* reduced the amount of insoluble protein during drought stress (figure 1) [22]. This function is further supported by the observation of decreased insoluble protein and less oxidation of soluble proteins in *ATG18a*-overexpressing apple lines [39].

rsob.royalsocietypublishing.org   Open Biol. **8**: 180162

Interestingly, overexpressing *ATG18a* in apple also improved the antioxidant system under drought stress. Increased activity of H$_2$O$_2$-scavenging enzymes and higher expression of genes involved in the ascorbate–glutathione (AsA–GSH) cycle, an H$_2$O$_2$ scavenging system, were observed in *ATG18a*-overexpressing apple compared to WT under drought stress [39]. An interesting question remaining is what mechanisms connect *ATG18a* overexpression to the improved antioxidant system.

## 4.3. Heat stress

*Arabidopsis atg5* and *atg7* mutants are hypersensitive to heat stress [114], suggesting that autophagy also functions during heat responses. In crop species, the induction of *ATG* genes and increased autophagosome formation under heat stress were observed in pepper and tomato [16,23,67]. Silencing *ATG5* or *ATG7* in tomato plants led to reduced induction of autophagy under heat stress, leading to compromised heat tolerance [23,67]. In addition, a natural thermotolerant pepper line was found to have higher autophagy activity compared to a thermosensitive pepper line [16]. Together, these results suggest that autophagy may confer heat tolerance on plants. Heat or heat-induced ROS can cause toxic effects such as protein aggregation [113]. It has been shown that more insoluble proteins accumulate in autophagy-defective plants in *Arabidopsis* and tomato during heat stress [23,114], indicating that autophagy may function to remove protein aggregates (figure 1).

In *Arabidopsis,* the transcription factor WRKY33 is required for heat tolerance [118]. Two homologues of *Arabidopsis* WRKY33 were identified in tomato, WRKY33a and WRKY33b [23]. Silencing of either *WRKY33a* or *WRKY33b* compromised heat tolerance, suggesting that they function under heat stress. The induction of the autophagy-related genes *ATG5*, *ATG7*, *NBR1a* and *NBR1b* by heat was impaired in *WRKY33a*- or *WRKY33b*-silenced plants, indicating that the WRKY33s s might be positive regulators of autophagy in tomato (figure 1) [23].

## 4.4. Endoplasmic reticulum stress

Various environmental stress conditions, such as salt and heat stress, can disrupt the protein folding pathway and cause the accumulation of unfolded and misfolded proteins in the ER lumen, termed ER stress [119]. In *Arabidopsis*, ER stress triggers activation of autophagy, dependent on inositol-requiring enzyme 1 (IRE1), an ER stress sensor [120,121]. Fragments of ER containing unfolded proteins are delivered to the vacuole by autophagosomes, suggesting that autophagy may function to degrade them during ER stress [120,121]. We recently demonstrated the induction of autophagy during ER stress in maize [49]. Tunicamycin, a chemical that disrupts protein folding and leads to ER stress, was applied to the roots of maize seedlings over a time course of up to 48 h. Analyses of gene expression and cellular events suggested a transition over time from adaptive activities to cell death under persistent ER stress. Autophagy was activated at both the pro-survival stage and the cell death stage [49]. The function of autophagy at the different stages and whether autophagy is involved in the transition from survival activities to cell death are interesting topics for future research.

# 5. Functions of autophagy in plant–microbe interactions

## 5.1. Plant–pathogen interactions

Autophagy functions in the response to biotic stress in crop species, and manipulation of autophagy alters disease resistance in several species. For example, in banana, inhibition of autophagy by the autophagy inhibitor 3-methyladenine (3-MA) compromised resistance to *Fusarium oxysporum f.* sp. *cubense* [15]. Silencing *ATG8* genes in cassava rendered plants more susceptible to *Xanthomonas axonopodis pv manihotis* (*Xam*) [24,43]. In wheat, *ATG8j*-silenced plants were more susceptible to the avirulent fungal pathogen *Puccinia striiformis f.* sp. *tritici* (*Pst*) race CRY23 [84]. Knocking down *ATG6* in wheat enhanced basal resistance to *Blumeria graminis f. sp. tritici* (*Bgt*) in a susceptible line, but compromised resistance in a resistant line carrying the resistance gene *Pm21* [82].

Several autophagy regulators that may function during biotic stress have been identified. In *Arabidopsis*, the transcription factor WRKY33 interacts with ATG18a and maintains the induction of *ATG18a* during *Botrytis* infection [122]. A recent study in cassava identified the related transcription factor WRKY20 as a possible regulator of autophagy during cassava bacterial blight [24]. WRKY20 expression was induced when plants were infected with the pathogen *Xanthomonas axonopodis pv manihotis* (*Xam*) and the WRKY20 protein was found to interact with ATG8a, ATG8f and ATG8h. WRKY20 also bound to the W-box of the *ATG8a* promoter and activated its transcription. *WRKY20*-silenced plants were more susceptible to disease, suggesting that WRKY20 is necessary for disease resistance [24].

A set of glyceraldehyde-3-phosphate dehydrogenases (GAPDHs) were shown to negatively regulate disease resistance in *Arabidopsis*, *Nicotiana benthamiana* and cassava, as indicated by the enhanced disease resistance in plants when cytosolic glycolytic GAPDHs (GAPCs) were silenced or knocked out [43,123,124]. GAPDHs regulate autophagy, and in *Arabidopsis*, knockout mutants of chloroplastic photosynthetic GAPDH1 (GAPA1) and GAPC1 exhibited constitutive autophagy [123]. In *Nicotiana benthamiana*, silencing of any one or all of the three *GAPC*s activated autophagy, and overexpressing *GAPC1* or *GAPC2* inhibited autophagy [124]. In cassava, silencing of all of the *GAPC*s also led to activation of autophagy. Interestingly, GAPC was found to interact with ATG3 in *Nicotiana benthamiana* and this interaction was inhibited by ROS, indicating the possibility that ROS may regulate autophagy through GAPCs [124]. In cassava, GAPC4 and GAPC6 also interacted with ATG8b and ATG8e [43]. Additionally, silencing ATG8b and ATG8e in either WT or GAPC-silenced cassava decreased disease resistance, suggesting that GAPCs may impact disease resistance by regulating autophagy [43]. However, more work is still needed to fully understand the relationship between plant defence and regulation of autophagy by GAPCs.

Emerging evidence suggests that autophagy can play both anti-microbial and pro-microbial roles in plant–pathogen interactions [125]. Autophagy plays a positive role in resistance to necrotrophic pathogens [126], and autophagy-defective mutants in *Arabidopsis* are more susceptible to these pathogens [30,122,127]. A recent study also showed that chemically inhibiting autophagy in tomato restored its susceptibility to a

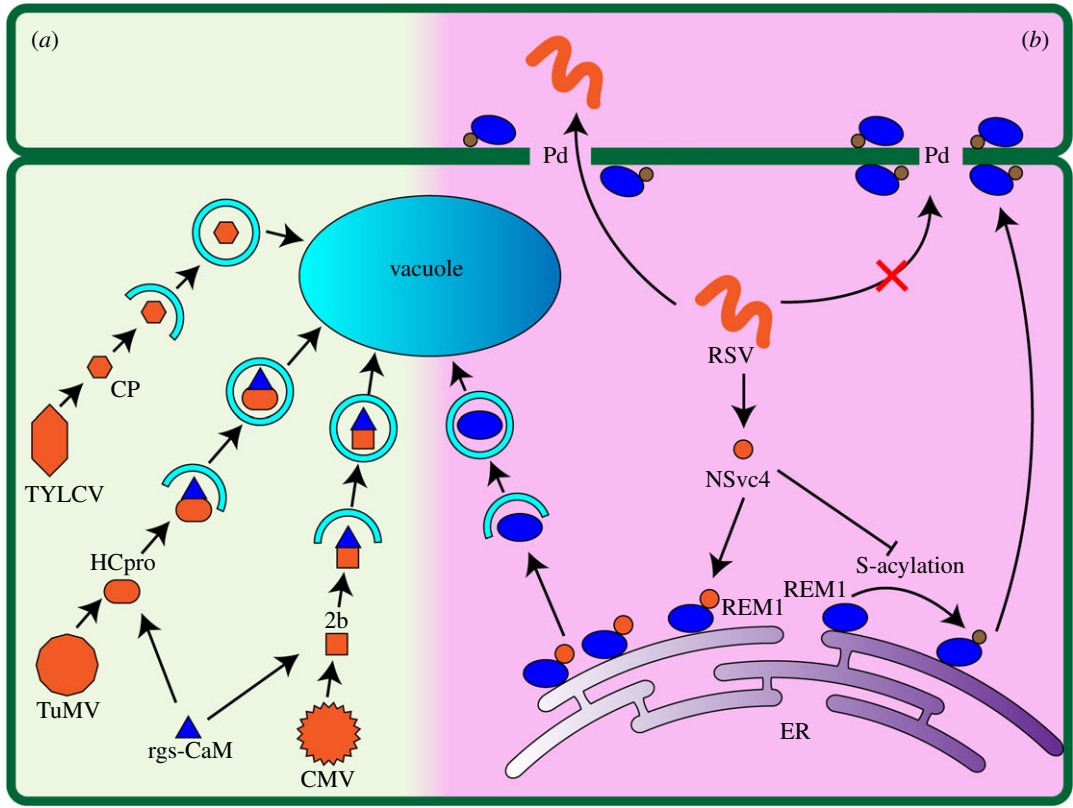

**Figure 2.** The dual role of autophagy during plant–pathogen interactions in crops. (*a*) Autophagy can play an anti-microbial role. Plant viruses express RNA silencing suppressors (RSSs) to inhibit the host RNA silencing pathway, such as the HCpro protein produced by turnip mosaic virus (TuMV) and 2b protein produced by cucumber mosaic virus (CMV). In *Nicotiana tabacum*, a calmodulin-like protein, rgs-CaM, can detect and bind to RSSs, preventing them from suppressing the host RNA silencing mechanism and promoting their degradation by autophagy. Meanwhile, rgs-CaM is degraded along with the RSSs. In tomato, autophagy is involved in degrading the coat protein (CP) of tomato yellow leaf curl virus (TYLCV). (*b*) Autophagy can play a pro-microbial role. In rice, group 1 remorin (REM1) undergoes S-acylation and is located in the plasma membrane and plasmodesmata (Pd), inhibiting the cell-to-cell movement of viruses. rice stripe virus (RSV) expresses a protein called NSvc4 that can bind to REM1, block its S-acylation, and retain REM1 in the ER. Decreased REM1 at the plasmodesmata enables RSV to move to another cell. The accumulation of non-acylated REM1 at the ER finally triggers autophagy for degradation.

non-pathogenic mutant strain of necrotrophic *Sclerotinia sclerotiorum* [68]. Interestingly, compared to the necrotic cell death triggered by the WT pathogen strain, this non-pathogenic mutant strain caused autophagic cell death in a more restricted area. This phenotype difference was later attributed to the lack of an autophagy suppressor in the non-pathogenic mutant strain [68], suggesting that autophagy functions to inhibit pathogen-induced cell death. When tomato was infected by tobacco mosaic virus, PCD was triggered in distal non-infected areas like the root tip [128], and autophagy was also activated by virus-induced ROS production in the root tip [69]. Inhibition of autophagy led to higher ROS accumulation, suggesting that autophagy contributes to balancing ROS levels and maintaining cell survival [69]. Several examples suggest that autophagy functions as an antiviral mechanism as well. In *Nicotiana benthamiana*, autophagy degrades the virulence factor βC1 of cotton leaf curl Multan virus through an interaction between ATG8 and βC1 [129]. In *Nicotiana tabacum*, the calmodulin-like protein rgs-CaM binds to viral RNA silencing suppressors (RSSs) and RSSs and rgs-CaM are degraded by autophagy (figure 2) [63]. In tomato, autophagy is one of the pathways that degrades the coat protein of tomato yellow leaf curl virus (figure 2) [70].

In turn, pathogens can overcome or even hijack host autophagy for their benefit. For example, in *Arabidopsis* and tomato, the necrotrophic fungal pathogen *Sclerotinia sclerotiorum* suppresses autophagy through the secretion of a phytotoxin called oxalic acid (OA) [68]. The expression of *ATG4*, *ATG8f* and *ATG8g* is significantly reduced after inoculation with the WT pathogen. An OA-defective mutant strain is non-pathogenic, but the pathogenicity is rescued by inhibiting autophagy in the host plants [68], further supporting the hypothesis that OA can suppress autophagy. An oomycete pathogen *Phytophthora infestans*, which causes late blight of potatoes and was responsible for the Irish potato famine in the nineteenth century, can also modulate autophagy through its effector PexRD54 [130]. PexRD54 can interact with ATG8 and co-localizes with ATG8 on autophagosomes. This interaction has been demonstrated to compete with the interaction between ATG8 and a cargo receptor for selective autophagy called Joka2, causing hypersensitivity to infection in plants [130].

Viruses are also able to suppress or hijack host autophagy to facilitate infection. In *Arabidopsis* and *Nicotiana benthamiana*, viruses were found to degrade components of the host antiviral RNA silencing system by hijacking autophagy [131–133] or to directly inhibit host autophagy by disrupting the ATG7–ATG8 interaction [134]. Remorins (REMs) are plasma membrane and plasmodesmatal proteins [135], and the group 1 remorins (REM1) from *Nicotiana benthamiana* and rice are targeted by rice stripe virus through hijacking of host autophagy (figure 2) [54], leading to a decrease in REM1. One biological function of REMs is inhibition of the cell-to-cell movement of viruses by binding to viral

rsob.royalsocietypublishing.org    *Open Biol.* **8**: 180162

**Table 4.** Findings on autophagy in crops compared to findings on autophagy in *Arabidopsis*.

| similarities with *Arabidopsis* | new findings in crop species |
| --- | --- |
| 1. Core *ATG* genes are present. | 1. Autophagy is involved in reproductive development (anther development in rice and tomato, spikelet development in wheat) and phloem development (in wheat). |
| 2. Autophagy-defective plants display characteristic conserved phenotypes (early senescence, hypersensitivity to stress etc.). | 2. Several new regulators of autophagy under abiotic and biotic stresses have been identified in tomato and cassava. |
| 3. Autophagy is involved in multiple developmental processes including leaf senescence, seed development and xylem development. | 3. Autophagy may play a role in symbiotic interactions in common bean, especially nodule development. |
| 4. Autophagy functions in abiotic and biotic stress responses. | |

movement proteins [135]. S-acylation of REM1 at the C-terminus is required for its localization to the plasma membrane and its stability; otherwise, REM1 accumulates in the ER and is degraded by autophagy. A viral protein, NSvc4, interacts with REM1 at its C-terminus, possibly blocking the S-acylation site, and impairs the S-acylation, leading to the degradation of REM1 by autophagy [54].

## 5.2. Symbiotic interactions

Research with the common bean (*Phaseolus vulgaris*) indicates that autophagy may be involved in symbiotic interactions. During the rhizobium–legume symbiotic interaction, trehalose is one of the greatly induced metabolites [136]. Silencing of trehalase, an enzyme required for trehalose degradation, leads to higher trehalose content, with increased bacterial viability, nodule biomass and N assimilation. The expression of *ATG3* is also increased in the nodules of the *TRE1*-RNAi plants [44], suggestive of a role for autophagy. The accumulation of trehalose was reported to trigger autophagy in a resurrection grass species, *Tripogon loliiformis*, during dehydration [137]; whether trehalose induces autophagy also in nodules and how autophagy contributes to symbiosis are interesting directions for future research.

A recent study showed that the transcript level of the autophagy-related gene *PI3K* was upregulated in the root hair and rhizobial entry site after the onset of nodule development [45]. In *PI3K*-RNAi plants, two typical processes during nodulation, root hair curling and infection thread (IT) formation, were negatively affected. Nodule primordia and nodules were decreased in number and smaller in size in *PI3K*-RNAi plants, and this was also observed in *ATG6*-RNAi plants [45]. Another typical type of symbiosis, colonization with arbuscular mycorrhizal fungi, was also impaired in *PI3K*-RNAi plants [45]. However, as the PI3K complex functions in multiple biological processes, these results are not sufficient to conclude that autophagy directly contributes to symbiotic interactions. In addition, decreasing the expression of target of rapamycin (TOR), a negative regulator of autophagy [138], using RNAi also impaired the infection process and altered nodule morphology [46]. Similar to the PI3K complex, TOR is involved in multiple biological processes besides autophagy. Thus, further evidence is needed to confirm whether the impaired symbiosis in *PI3K*-RNAi plants and *TOR*-RNAi plants is directly related

to autophagy. Characterizing plants after silencing other *ATG* genes may help to answer this question. Although both *PI3K*-RNAi plants and *TOR*-RNAi plants showed impaired symbiosis, autophagy should be reduced in *PI3K*-RNAi plants and activated in *TOR*-RNAi plants, as suggested by the decreased *ATG* transcript levels in *PI3K*-RNAi plants [45] and by an increase in both *ATG* transcripts and autophagosome number in *TOR*-RNAi plants [46]. If autophagy is indeed directly involved in symbiosis, an outstanding question would be whether maintaining autophagy at a certain level is required for successful symbiotic interactions.

## 6. Future perspectives

The study of autophagy in crop species has been expanding rapidly. Functions of autophagy in development, abiotic stress responses and plant–microbe interactions have been deciphered in various species. New findings such as the involvement of autophagy in reproductive development [34,53,65,79] are increasing our understanding of autophagy (table 4), but much work is still needed. One interesting topic that warrants more attention is the role of autophagy in organs or tissues that are specifically present in certain crops, for example, fruits and nodules.

Several transcriptional or post-translational regulators of autophagy have been identified and characterized in crops. The identification and characterization of new regulatory mechanisms is a critical area for future research. Some important regulators characterized in *Arabidopsis* have not yet been well-studied in crops, for example, TOR [138] and Snf1-related protein kinase 1 (SnRK1) [139]. While the mechanism of regulation of autophagy has been considered to be primarily post-translational, there are now a number of examples indicating that the expression of *ATG* genes can change in response to developmental processes and environmental cues. The transcriptional control of autophagy should be another fruitful area for further research.

Considering its importance in development and stress responses, autophagy is a promising target to manipulate for agricultural benefits like higher yield. Increased expression of *ATG* genes may be valuable in agricultural applications, as this can confer a number of benefits to plants, including enhanced growth, higher yield and increased stress tolerance [12,38,39].

Data accessibility. This article has no additional data.

Authors' contributions. J.T. and D.C.B. outlined the article; J.T. drafted the article; D.C.B. revised the article. All authors gave final approval for publication.

Competing interests. We declare we have no competing interests.

Funding. This work was supported by grant no. IOS 1444339 from the National Science Foundation Plant Genome Research Program to D.C.B.

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
