## [Reviewer comments · Open Biology]

Review History

RSOB-18-0162.R0 (Original submission)

Review form: Reviewer 1

Recommendation

Accept with minor revision (please list in comments)

Are each of the following suitable for general readers?

- a) **Title**
Yes
- b) **Summary**
Yes
- c) **Introduction**
Yes

Is the length of the paper justified?

Yes

Should the paper be seen by a specialist statistical reviewer?

No

Is it clear how to make all supporting data available?

Not Applicable

Is the supplementary material necessary; and if so is it adequate and clear?

Not Applicable

Do you have any ethical concerns with this paper?

No

Comments to the Author

Minor comments:

1. Introduction chapter. Typically ATG8-lipidation machinery is referred to as two ubiquitin-like conjugation systems.
2. While mentioning the studies involving over-expression or silencing of individual ATG genes authors do not dwell on the correlation between expression level of these genes and actual activity of the pathway. It is especially misleading in the context of the previous chapter mentioning transcriptional changes of endogenous genes under autophagy-inducing conditions.
3. It would be interesting to also know authors opinion on autophagy-proteasome interplay in homeostasis.
4. The evidence for existence of microautophagy in plants is still questionable. For example, conclusions in the doi: 10.1002/cbin.10512 cited in this review, are based on electron microscopy that does not exclude the role of macroautophagy.

Review form: Reviewer 2

Recommendation

Major revision is needed (please make suggestions in comments)

Are each of the following suitable for general readers?

- a) **Title**
Yes
- b) **Summary**
Yes
- c) **Introduction**
Yes

Is the length of the paper justified?

Yes

Should the paper be seen by a specialist statistical reviewer?

No

Is it clear how to make all supporting data available?

No

Is the supplementary material necessary; and if so is it adequate and clear?

No

Do you have any ethical concerns with this paper?

No

Comments to the Author

The present review manuscript shows the recent progress on understanding autophagy in crops as well as other plant species and discuss potential future research directions. I think this topic itself is timely and highly valuable for general readers of "Open Biology". The authors have covered much of the relevant literature. However, I think there are some problems with the manuscript. The following are some of the critical comments.

(1) Figure 1: similar figures have already been published in many reviews and not new. Not worth to be published repeatedly here in this review. No need to publish it here.

(2) Table 1: The authors found new ATG genes from some plant species by using the database search, and these genes have been mentioned in Table 1. However, the readers can't get the information about these genes. The authors should mention the accession codes of these genes and show the similarity data (e.g. Amino acids) compared with Arabidopsis ATGs.

(3) Table 2: It is hard to understand the relationships between "Related processes" and "References" for general readers. The authors should accurately describe relevant processes and corresponding citations.

(4) Figure 2: Title in figure 2 that "regulation of autophagy-----in crops" is too speculative and NOT justified conclusively. The title should be revised in a more realistic manner based upon current results (only in Tomato and Arabidopsis?).

(5) Title that "Autophagy in Crop Plants-What's New Beyond Arabidopsis?" contains interesting theme for general readers as well as plant autophagy researchers. However, the authors do not provide clear explanation about this theme in all figures and tables. At present, it is hard to understand "What's New Beyond Arabidopsis?" from all figures and tables. Can the authors propose a model/table about "What is new findings from the study of crops?" and "What is consistent with the results of Arabidopsis?"

Decision letter (RSOB-18-0162.R0)

11-Oct-2018

Dear Dr Bassham

We are pleased to inform you that your manuscript RSOB-18-0162 entitled "Autophagy in Crop Plants - What's New Beyond Arabidopsis?" has been accepted by the Editor for publication in Open Biology. The reviewer(s) have recommended publication, but also suggest some minor

revisions to your manuscript. Therefore, we invite you to respond to the reviewer(s)' comments and revise your manuscript.

Please submit the revised version of your manuscript within 14 days. If you do not think you will be able to meet this date please let us know immediately and we can extend this deadline for you.

- 1) A text file of the manuscript (doc, txt, rtf or tex), including the references, tables (including captions) and figure captions. Please remove any tracked changes from the text before submission. PDF files are not an accepted format for the "Main Document".
- 2) A separate electronic file of each figure (tiff, EPS or print-quality PDF preferred). The format should be produced directly from original creation package, or original software format. Please note that PowerPoint files are not accepted.
- 3) Electronic supplementary material: this should be contained in a separate file from the main text and meet our ESM criteria (see <http://royalsocietypublishing.org/instructions-authors#question5>). All supplementary materials accompanying an accepted article will be treated as in their final form. They will be published alongside the paper on the journal website and posted on the online figshare repository. Files on figshare will be made available approximately one week before the accompanying article so that the supplementary material can be attributed a unique DOI.

Online supplementary material will also carry the title and description provided during submission, so please ensure these are accurate and informative. Note that the Royal Society will not edit or typeset supplementary material and it will be hosted as provided. Please ensure that the supplementary material includes the paper details (authors, title, journal name, article DOI). Your article DOI will be 10.1098/rsob.2016[last 4 digits of e.g. 10.1098/rsob.20160049].

- 4) A media summary: a short non-technical summary (up to 100 words) of the key findings/importance of your manuscript. Please try to write in simple English, avoid jargon, explain the importance of the topic, outline the main implications and describe why this topic is newsworthy.

Images

Data-Sharing

It is a condition of publication that data supporting your paper are made available. Data should be made available either in the electronic supplementary material or through an appropriate repository. Details of how to access data should be included in your paper. Please see <http://royalsocietypublishing.org/site/authors/policy.xhtml#question6> for more details.

Data accessibility section

Sincerely,

The Open Biology Team
<mailto:openbiology@royalsociety.org>

ditage Insights by clicking on the following link: <https://www.surveymonkey.com/r/author-perspectives-on-academic-publishing-royal-society>

This should take no more than 15 minutes and you will have the opportunity to enter a prize draw. We hope these results will provide us with valuable insights we can use to improve our service.

Reviewer(s)' Comments to Author:

Referee: 1

Comments to the Author(s)

Minor comments:

1. Introduction chapter. Typically ATG8-lipidation machinery is referred to as two ubiquitin-like conjugation systems.
2. While mentioning the studies involving over-expression or silencing of individual ATG genes authors do not dwell on the correlation between expression level of these genes and actual activity of the pathway. It is especially misleading in the context of the previous chapter mentioning transcriptional changes of endogenous genes under autophagy-inducing conditions.
3. It would be interesting to also know authors opinion on autophagy-proteasome interplay in homeostasis.
4. The evidence for existence of microautophagy in plants is still questionable. For example, conclusions in the doi: 10.1002/cbin.10512 cited in this review, are based on electron microscopy that does not exclude the role of macroautophagy.

Referee: 2

Comments to the Author(s)

The present review manuscript shows the recent progress on understanding autophagy in crops as well as other plant species and discuss potential future research directions. I think this topic itself is timely and highly valuable for general readers of "Open Biology". The authors have covered much of the relevant literature. However, I think there are some problems with the manuscript. The following are some of the critical comments.

(1) Figure 1: similar figures have already been published in many reviews and not new. Not worth to be published repeatedly here in this review. No need to publish it here.

(2) Table 1: The authors found new ATG genes from some plant species by using the database search, and these genes have been mentioned in Table 1. However, the readers can't get the information about these genes. The authors should mention the accession codes of these genes and show the similarity data (e.g. Amino acids) compared with Arabidopsis ATGs.

(3) Table 2: It is hard to understand the relationships between "Related processes" and "References" for general readers. The authors should accurately describe relevant processes and corresponding citations.

(4) Figure 2: Title in figure 2 that "regulation of autophagy-----in crops" is too speculative and NOT justified conclusively. The title should be revised in a more realistic manner based upon current results (only in Tomato and Arabidopsis?).

(5) Title that "Autophagy in Crop Plants-What's New Beyond Arabidopsis?" contains interesting theme for general readers as well as plant autophagy researchers. However, the authors do not provide clear explanation about this theme in all figures and tables. At present, it is hard to understand "What's New Beyond Arabidopsis?" from all figures and tables. Can the authors propose a model/table about "What is new findings from the study of crops?" and "What is consistent with the results of Arabidopsis?"

Author's Response to Decision Letter for (RSOB-18-0162.R0)

See Appendix A.

Decision letter (RSOB-18-0162.R1)

08-Nov-2018

Dear Dr Bassham,

We are pleased to inform you that your manuscript entitled "Autophagy in Crop Plants – What's New Beyond Arabidopsis?" has been accepted by the Editor for publication in Open Biology.

You can expect to receive a proof of your article from our Production office in due course, please

check your spam filter if you do not receive it within the next 10 working days. Please let us know if you are likely to be away from e-mail contact during this time.

Sincerely,

The Open Biology Team
mailto: openbiology@royalsociety.org

Decision letter (RSOB-18-0162.R2)

08-Nov-2018

Dear Dr Bassham

We are pleased to inform you that your manuscript entitled "Autophagy in Crop Plants – What's New Beyond Arabidopsis?" has been accepted by the Editor for publication in Open Biology.

Sincerely,

The Open Biology Team
mailto: openbiology@royalsociety.org

Appendix A

Response to reviewers

Referee: 1

Comments to the Author(s)

Minor comments:

1. Introduction chapter. Typically ATG8-lipidation machinery is referred to as two ubiquitin-like conjugation systems.

We rewrote this part in the introduction (lines 46-48) and included information on the two ubiquitin-like conjugation systems.

2. While mentioning the studies involving over-expression or silencing of individual ATG genes authors do not dwell on the correlation between expression level of these genes and actual activity of the pathway. It is especially misleading in the context of the previous chapter mentioning transcriptional changes of endogenous genes under autophagy-inducing conditions.

We rewrote some sentences to clarify the connection between expression level of ATG genes and activity of pathways (autophagy or other affected pathways) as well as how the activity is altered in these plants under stress conditions (see lines 55-56, 224-227, 245-246, 300-302).

3. It would be interesting to also know authors opinion on autophagy-proteasome interplay in homeostasis.

We agree that this is an interesting topic. However, to our knowledge there is little information currently on this topic in crops, and therefore we have chosen not to speculate until more research is available.

4. The evidence for existence of microautophagy in plants is still questionable. For example, conclusions in the doi: 10.1002/cbin.10512 cited in this review, are based on electron microscopy that does not exclude the role of macroautophagy.

We agree that microautophagy is still not well-studied in plants and mention this point in the introduction. In addition, we included two more examples (uptake of anthocyanin and damaged chloroplasts by the vacuole) to support the existence of microautophagy (lines 25-27).

Referee: 2

Comments to the Author(s)

The present review manuscript shows the recent progress on understanding autophagy in crops as well as other plant species and discuss potential future research directions. I think this topic itself is timely and highly valuable for general readers of "Open Biology". The authors have covered much of the relevant literature. However, I think there are some problems with the manuscript. The following are some of the critical comments.

(1) Figure 1: similar figures have already been published in many reviews and not new. Not worth to be published repeatedly here in this review. No need to publish it here.

We deleted this figure. Correspondingly, the figure numbers of the other two figures were changed to figure 1 and figure 2 in the text and figure legends.

(2) Table 1: The authors found new ATG genes from some plant species by using the database search, and these genes have been mentioned in Table 1. However, the readers can't get the information about these genes. The authors should mention the accession codes of these genes and show the similarity data (e.g. Amino acids) compared with Arabidopsis ATGs.

We created a table for the newly identified genes that includes the accession numbers and similarity data compared to *Arabidopsis* ATGs (Table 2).

(3) Table 2: It is hard to understand the relationships between “Related processes” and “References” for general readers. The authors should accurately describe relevant processes and corresponding citations.

We changed the layout of this table and added references directly to the corresponding process. Note that this table is table 3 in the revised version.

(4) Figure 2: Title in figure 2 that “regulation of autophagy-----in crops” is too speculative and NOT justified conclusively. The title should be revised in a more realistic manner based upon current results (only in Tomato and *Arabidopsis*?).

We rewrote the title to “Identified regulators of autophagy during drought and heat stress in tomato”. Note that this figure is figure 1 in the revised version.

(5) Title that “Autophagy in Crop Plants-What’s New Beyond *Arabidopsis*?” contains interesting theme for general readers as well as plant autophagy researchers. However, the authors do not provide clear explanation about this theme in all figures and tables. At present, it is hard to understand “What’s New Beyond *Arabidopsis*?” from all figures and tables. Can the authors propose a model/table about “What is new findings from the study of crops?” and “What is consistent with the results of *Arabidopsis*?”

We created Table 4 as a general summary of consistent findings and new findings in crops compared to what has been found in *Arabidopsis*.